# Hormonal Regulation of Glucocorticoid Inactivation and Reactivation in αT3-1 and LβT2 Gonadotroph Cells

**DOI:** 10.3390/biology8040081

**Published:** 2019-10-26

**Authors:** Anthony E. Michael, Lisa M. Thurston, Robert C. Fowkes

**Affiliations:** 1Biological & Chemical Sciences, Queen Mary, University of London, Queen Mary, University of London, Mile End Road, London E1 4NS, UK; a.michael@qmul.ac.uk; 2Department of Comparative Biomedical Sciences, Royal Veterinary College, Royal College Street, Camden, London NW1 0TU, UK; lthurston@rvc.ac.uk; 3Endocrine Signalling Group, Royal Veterinary College, Royal College Street, Camden, London NW1 0TU, UK

**Keywords:** cortisol, glucocorticoid, pituitary, GnRH, PACAP, gonadotroph, hydroxysteroid dehydrogenase

## Abstract

The regulation of reproductive function by glucocorticoids occurs at all levels of the hypothalamo-pituitary-gonadal axis. Within the pituitary, glucocorticoids have been shown to directly alter gene expression in gonadotrophs, indicating that these cell types are sensitive to regulation by the glucocorticoid receptor. Whilst the major glucocorticoid metabolising enzymes, 11β-hydroxysteroid dehydrogenase (11βHSD; HSD11B1 and HSD11B2), have been described in human pituitary adenomas, the activity of these enzymes within different pituitary cell types has not been reported. Radiometric conversion assays were performed in αT3-1, LβT2 (gonadotrophs), AtT-20 (corticotrophs) and GH3 (somatolactotrophs) anterior pituitary cell lines, using tritiated cortisol, corticosterone, cortisone or 11-dehydrocorticosterone as substrates. The net oxidation of cortisol/corticosterone and net reduction of cortisone/11-dehydrocorticosterone were significantly higher in the two gonadotroph cells lines compared with the AtT-20 and GH3 cells after 4 h. Whilst these enzyme activities remained the same in αT3-1 and LβT2 cells over a 24 h period, there was a significant increase in glucocorticoid metabolism in both AtT-20 and GH3 cells over this same period, suggesting cell-type specific activity of the 11βHSD enzyme(s). Stimulation of both gonadotroph cell lines with either 100 nM GnRH or PACAP (known physiological regulators of gonadotrophs) resulted in significantly increased 11β-dehydrogenase (11βDH) and 11-ketosteroid reductase (11KSR) activities, over both 4 and 24 h. These data reveal that gonadotroph 11βHSD enzyme activity can act to regulate local glucocorticoid availability to mediate the influence of the HPA axis on gonadotroph function.

## 1. Introduction

The phenomenon of stress-related infertility, observed in many distinct species of mammals and lower vertebrates, reflects multiple endocrine interactions between the reproductive (hypothalamo-pituitary-gonadal) and stress (hypothalamo-pituitary-adrenal) axes. Several studies by independent research teams have established that both pharmacological and physiological glucocorticoids (corticosterone in myomorph rodents, i.e., rats and mice; cortisol in most other mammals) can exert direct effects to suppress reproduction at the level of the hypothalamus, anterior pituitary, gonads and reproductive tract [1,2,3,4,5].

Throughout the body, the ability of the physiological glucocorticoids to access nuclear receptors and hence exert cellular actions are modulated by the 11β-hydroxysteroid dehydrogenase (11βHSD) isoenzymes. These members of the short-chain alcohol dehydrogenase enzyme superfamily catalyze the inter-conversion of active glucocorticoids (cortisol and corticosterone) with their inert 11-ketometabolites (cortisone and 11-dehydro-corticosterone, respectively) [6,7,8,9,10]. The first cloned isoenzyme, type 1 11βHSD (HSD11B1), is a bidirectional, low-affinity enzyme that predominantly catalyzes the reduction of cortisone to cortisol (and 11-dehydrocorticosterone to corticosterone) using reduced nicotinamide adenine dinucleotide phosphate (NADPH) as the preferred pyridine dinucleotide cosubstrate [8,9,10]. However, the direction of HSD11B1 activity is determined by the redox state of NADP+/NADPH in the lumen of the smooth endoplasmic reticulum that is, in turn, dependent upon the activity of the microsomal hexose-6-phosphate dehydrogenase enzyme (H6PDH) [11,12,13,14,15,16,17]. Hence, under defined circumstances, HSD11B1 can act as a low affinity, dehydrogenase enzyme to catalyze the NADP+-dependent inactivation of physiological glucocorticoids [6,7,8,9,10]. In contrast to the bidirectional HSD11B1 enzyme, the type 2 11βHSD (HSD11B2) isoenzyme is a high-affinity enzyme that acts exclusively as a dehydrogenase to catalyze the inactivation of cortisol (and corticosterone) using oxidized nicotinamide adenine dinucleotide (NAD+) as an obligate cosubstrate [6,7,10]. 

Hong et al. [18] have recently described the effects of the endocrine-disrupting chemical, mono-(2-ethylhexyl) phthalate (MEHP), on the activity of HSD11B2 in LβT2 gonadotroph cells. Whilst Korbonits et al. [19] have attempted to characterize the expression of both cloned 11βHSD enzymes in a range human pituitary adenomas of various origins, there is a paucity of data regarding possible expression of HSD11B1 in gonadotrophs. Furthermore, despite gonadotrophs being established anterior pituitary target cells for glucocorticoids [20,21,22,23,24,25,26], there are no published studies into the hormonal regulation of glucocorticoid metabolism in gonadotroph cells. Hence, the objectives of the current study were to assess glucocorticoid metabolism in well-characterized anterior pituitary cell lines and to evaluate the regulation of steroid metabolism in those gonadotroph cell lines by two physiological stimuli, which were the gonadotrophin-releasing hormone (GnRH) and pituitary adenylate cyclase-activating polypeptide (PACAP).

## 2. Materials and Methods

### 2.1. Materials

All chemicals were purchased from Sigma (Poole, UK) unless otherwise stated. PACAP 1-38 was obtained from Merck (Nottingham, UK) in lyophilized form and stored at −20 °C as stock solutions of 1mM in sterile water.

### 2.2. Cell Culture

αT3-1, LβT2, AtT-20 and GH3 cells were each grown in monolayer cultured in Dulbecco’s Modified Eagle medium (DMEM) supplemented with high glucose (4500 mg/L), 10% (v/v) fetal calf serum (FCS), penicillin (100 IU/mL), streptomycin (100 mg/L) and fungizone (125 mg/L) (Invitrogen, Paisley, UK) (hereafter referred to as culture medium), as described previously [27]. Cells were passaged twice weekly and incubated at 37 °C in a humidified atmosphere of 5% (v/v) CO2 in air. For radiometric conversion assays, cells were sub-cultured in serum-free culture medium in 24-well plates at a density of 3 × 10^5^ cells/well. For mRNA extraction and PCR, cells were sub-cultured in a serum-supplemented cultured medium in 6-well plates at a density of 1 × 10^6^ cells/well.

### 2.3. Radiometric Conversion Assays

In order to assess the enzyme-mediated inter-conversion of active cortisol and inert cortisone by the 11βHSD isoenzymes in intact cells, each of the anterior pituitary cell lines was sub-cultured for either 4 h or 24 h in serum-free culture medium supplemented with 100 nmol/L of [1,2,6,7-^3^H]-cortisol (GE Healthcare, Amersham, Bucks., UK), [1,2,6,7-^3^H]-corticosterone (GE Healthcare, Amersham, Bucks., UK), [1,2,6,7-^3^H]-cortisone or [1,2,6,7-^3^H]-11-dehydrocorticosterone (specific activity of each adjusted using non-radioactive steroids to 18.5 kBq/100 pmol). At the end of the incubation, 1 mL volume of the supernatant medium containing the radio-labeled steroids was transferred from each of the wells of the culture plate into separate screw-capped borosilicate tubes whereupon 2 mL chloroform was added to each tube to effect the organic extraction of the radio-labeled steroid substrate and metabolites. After centrifugation at 1000 × g for 30 min at 4 °C, the aqueous phase was aspirated and discarded as radioactive waste, before then evaporating the organic steroid extract to dryness under a stream of inert nitrogen gas at a temperature of 45 °C. The concentrated steroid extracts were then re-suspended in 20 μL volumes of ethyl acetate loaded with 1 mmol/L non-radioactive cortisol and 1 mmol/L non-radioactive cortisone before being transferred onto the separate lanes on a Silica 60 TLC plate. Thereafter, the TLC plates were developed in a humidified atmosphere of 92:8 chloroform:95% (v/v) ethanol and transferred onto an AR200 Bioscan radiochromatogramme where inline Laura Lite software (v3.0) (LabLogic, Sheffield, UK) could be used to quantify the fractional conversion of the [^3^H]-steroid substrates to the corresponding [^3^H]-steroid products. To calculate the rate of product generation in pmol product, it was necessary to multiply the fractional conversion of substrates by the absolute amount of each steroid. For the assays of net 11-ketosteroid reductase activities, [^3^H]-cortisone and [^3^H]-11-dehydro-corticosterone were generated in house by incubating homogenates of male rat kidney overnight at 37 °C with 740 kBq of [1,2,6,7-^3^H]-cortisol or [1,2,6,7-^3^H]-corticosterone per tube plus 400 μmol/L NADP+ as oxidative cosubstrate for HSD11B1, as previously described [28]. After resolving the [^3^H]-cortisone / [^3^H]-11-dehydrocorticosterone, as appropriate, by TLC, the area containing the radioactive 11-ketosteroid was scraped from the TLC plates, transferred into a borosilicate tube and extracted overnight into 5 mL of n-hexane. The silica slurry was then sedimented by centrifugation (1000 × g at 4 °C for 30 min), the supernatant organic solvent was transferred to a second borosilicate tube and the slurry was extracted for a second time into 5mL n-hexane. After precipitating the second batch of silica slurry, the n-hexane was pooled with the prior extract and the entire volume was evaporated at 45 °C under nitrogen before being resuspended in a minimal volume of 9:1 (v/v) toluene:ethanol. This method recovered 87.6 ± 1.0% of the initial amount of radioactivity from each tube, and generated [^3^H]-11-keto-steroids which were 96% pure (with no single contaminant using the same chloroform:ethanol TLC solvent system as that used for all of the radiometric conversion assays). All radiometric conversion assays were repeated 3 times using cells from a different passage number on each occasion.

### 2.4. Data Presentation and Analyses

Experiments were performed from three different passages of cells, each performed in triplicate. Pooled data represent the mean ± SEM of these three independent experiments and were subjected to ANOVA followed by either Tukey–Kramer or Dunnet’s multiple comparisons tests (accepting *p* < 0.05 as significant), using in-built equations in GraphPad Prism 7.0a (GraphPad, San Diego, CA, USA). In some instances, data are presented as pooled, normalised to enzyme activity of untreated cells within each individual experiment and shown as % of control.

## 3. Results

### 3.1. Comparative 11βHSD Activities in Pituitary Cell Lines

Both 11β-dehydrogenase and 11-ketosteroid reductase activities could be measured in all four anterior pituitary cell lines irrespective of the radiometric conversion assay duration (both at 4 and 24 h) or the steroid substrate (cortisol/cortisone or corticosterone/11-dehydrocorticosterone) (Figure 1 and Figure 2). Over a 4 h incubation with steroid substrate, the net oxidation of 100 nM cortisol/corticosterone and net reduction of 100 nM cortisone/11-dehydrocorticosterone tended to be significantly higher in the two murine gonadotroph cells lines compared with the AtT-20 corticotrophs and GH3 somatotrophs (*p* < 0.05; Figure 1A and Figure 2A). However, in the αT3-1 and LβT2 cells, levels of steroid metabolism at 24 h did not differ significantly from those observed after 4 h, whereas the amount of product formed after 24 h in both the AtT-20 and GH3 cells was up to 31.1-fold higher than the amount of the corresponding product generated over the first 4 h (Figure 1 and Figure 2). Consequently, following a 24 h incubation with 100 nM [^3^H]-steroid substrates, the net oxidation of cortisol/corticosterone or reduction of cortisone/11-dehydrocorticosterone were both significantly lower in the gonadotroph cell lines compared with the AtT-20 and GH3 cells (*p* < 0.05; Figure 1A,B). In most cases, the net levels of glucocorticoid metabolism were comparable in the two gonadotroph cell lines. However, after 4 h with corticosterone and 11-dehydrocorticosterone, and after a 24 h incubation with cortisone, there were significantly higher levels of steroid metabolism in the LβT2 cells than in the αT3-1 cells (*p* < 0.05 in each case; Figure 1 and Figure 2).

### 3.2. Regulation of Cortisol-Cortisone Inter-conversion by 11βHSD in Gonadotroph Cell Lines

Both GnRH and PACAP were capable of regulating 11βHSD activities over a 4 and/or 24 h incubation and in at least one of the two enzymatic directions (Table 1). Specifically, GnRH stimulated the inter-conversion of cortisol-cortisone in both gonadotroph cell lines (Figure 3 and Figure 4). In αT3-1 cells, GnRH stimulated both the net 11β-dehydrogenase and 11-ketosteroid reductase activities over both 4 and 24 h (*p* < 0.05; Figure 3). In contrast, in LβT2 cells, GnRH had no significant effect on either enzymatic activity over 4 h (Figure 4A), but stimulated both the oxidation of cortisol and the reduction of cortisone over 24 h (*p* < 0.05; Figure 4B).

With the exception of the stimulation of 11-ketosteroid reductase activities in αT3-1 cells, the effects of GnRH were generally replicated by PACAP (Table 1). This polypeptide stimulated the net oxidation of cortisol in both αT3-1 cells and LβT2 cells, both over 4 and 24 h (*p* < 0.01 in each case; Figure 3 and Figure 4), and also stimulated the 11-ketosteroid reductase activities in the LβT2 cells over 24 (but not 4) h (*p* < 0.01; Figure 4B).

## 4. Discussion

This study is the first to investigate the hormonal regulation of 11βHSD activities in intact gonadotroph cell lines, and to assess glucocorticoid metabolism by the 11βHSD enzymes in non-gonadotroph anterior pituitary cells (specifically in AtT-20 and GH3 cell lines). Whilst being more physiologically relevant, in using intact cells, it is impossible for us to ascribe the net oxidation of cortisol and corticosterone to their inert 11-ketosteroid metabolites to either of the two cloned 11βHSD isoenzymes. However, we would note that Hong et al. [18] reported on the expression and activity of HSD11B2 in LβT2 cells, while Korbonits et al. [19] had previously shown both HSD11B1 and HSD11B2 to be expressed in human anterior pituitary adenomas. 

While HSD11B2 has been reported to catalyze the reduction of 11-dehydrodexamethasone to dexamethasone, this NAD+-dependent, high-affinity isoenzyme acts exclusively as a 11β-dehydrogenase with physiological glucocorticoid substrates, i.e., this enzyme cannot reduce cortisone to cortisol or 11-dehydrocorticosterone to corticosterone [29,30]. Although it is not possible to ascribe the oxidative metabolism of cortisol or corticosterone unambiguously to a specific 11BHSD isoenzyme (as both cloned HSD11B enzymes can catalyze this reaction), the reduction of cortisone and 11-dehydocorticosterone to active physiological glucocorticoids has only ever been attributed to HSD11B1. (While HSD11B2 has been reported to reduce the synthetic substrate 11-dehydrodexamethasone to the pharmacological glucocorticoid dexamethasone [31], this enzyme cannot reduce physiological 11-ketosteroid substrates.) Hence, it is reasonable to attribute the reduction of cortisone and 11-dehydrocorticosterone to active glucocorticoid metabolites to the HSD11B1 enzyme. Accepting that targeted knockdown of HSD11B2 expression would provide definitive proof that steroid metabolism was being mediated in the pituitary cell lines specifically by HSD11B1, this would seem a reasonable inference based on our observations of bidirectional steroid metabolism reported herein.

One of the major findings of the comparative study was that the net metabolism of steroid substrates in the two non-gonadotroph cell lines was between 7- and 31-fold higher after 24 h than that which was observed in the corresponding cell lines after just 4 h. This raises the possibility that during the course of the 24 h incubation with 100 nM steroid substrates in the AtT-20 and GH3 cells, there was steroidal induction of enzyme activities (which would almost certainly reflect the upregulation of enzyme expression). The fact that similar, non-linear increases were observed for both the net oxidation of cortisol/corticosterone and the net reduction of cortisone/11-dehydrocorticosterone would suggest induction of the bidirectional HSD11B1 enzyme. It is relevant to note that others have previously reported on the upregulation of HSD11B1 expression induced by glucocorticoids [32,33,34,35,36,37,38,39,40,41,42,43,44]. Hence, in the current study, cortisol/corticosterone may have served simultaneously as hormonal regulators of enzyme expression and/or activity as well as enzyme substrates.

Within the two gonadotroph cell lines, there were several cases where the rates of glucocorticoid metabolism were significantly higher in the LβT2 cells than in the αT3-1 cells. Specifically, the levels of corticosterone/11-dehydrocorticosterone inter-conversion over 4 h, and the net reduction of cortisone to cortisol over 24 h were each significantly higher in the LβT2 cells as compared to the αT3-1 cell line. It would appear, therefore, that the rates of 11βHSD-mediated glucocorticoid metabolism might be increased in the more mature/differentiated gonadotroph cell line, capable of expressing the specific LHβ subunit as compared to the relatively immature, undifferentiated αT3-1 cells that only express the common gonadotrophin α-subunit [45,46].

Turning to the endocrine regulation of enzyme activities, while this paper reports the first study of the hormonal regulation of glucocorticoid metabolism in anterior pituitary cells, there are previously published studies that suggest that HSD11B1 and/or HSD11B2 may be amenable to regulation by the second messenger and intracellular signaling systems activated by GnRH and PACAP. For example, the activities of both HSD11B enzymes are known to be regulated by calcium and by protein kinase C [47,48,49,50,51,52] while cyclic adenosine-3’,5’-monophosphate (cAMP) has been reported to stimulate glucocorticoid inactivation by HSD11B2 enzymes in placental [52,53,54] and renal cells [48,49], whereas we have previously shown cAMP/PKA to inhibit 11-ketosteroid reductase activity in boar testis [55].

The ability of GnRH and PACAP to increase significantly the net oxidation of cortisol/corticosterone and the reduction of cortisone/11-dehydrocorticosterone indicates that these two stimuli are likely to have stimulated the bidirectional HSD11B1 enzyme, although we cannot exclude additional stimulation of the oxidative HSD11B2 isoenzyme. The fact that the effects of GnRH and PACAP on steroid metabolism were largely concordant in the two gonadotroph cell lines would suggest the involvement of common or convergent intracellular signaling pathways in gonadotrophs, although the elucidation of the downstream signaling elements fell outside the scope of our current study.

## 5. Conclusions

In conclusion, our data show that in four different pituitary cell lines, including two different gonadotroph and two non-gonadotroph cell lines, the local actions of glucocorticoids are subject to modulation by 11βHSD enzymes that can both increase and decrease the intracellular concentrations of active glucocorticoids. Moreover, in both αT3-1 and LβT2 gonadotroph cells, the enzymatic inter-conversion of active glucocorticoids and their corresponding inert 11-ketosteroid metabolites can be stimulated by GnRH and by PACAP. Hence, these hormonal regulators of the HPG axis have the potential to moderate the direct effects of glucocorticoids, arising from the HPA axis, on anterior pituitary function.

## Figures and Tables

**Figure 1 biology-08-00081-f001:**
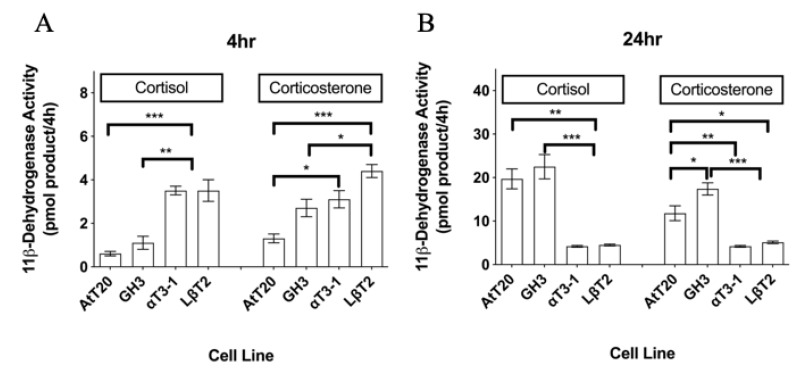
11β-dehydrogenase activities in four pituitary cell lines measured over (**A**) 4 h or (**B**) 24 h using 100 nM [^3^H]-cortisol and 100 nM [^3^H]-corticosterone as enzyme substrates. Data shown are mean ± SEM for three different cell passages pooled together (n = 3) (with triplicate measures in each passage). For a given time point and substrate, mean values with different superscripts differ significantly (*p* < 0.05) (one-way ANOVA with the Tukey–Kramer multiple comparison as the post hoc test).

**Figure 2 biology-08-00081-f002:**
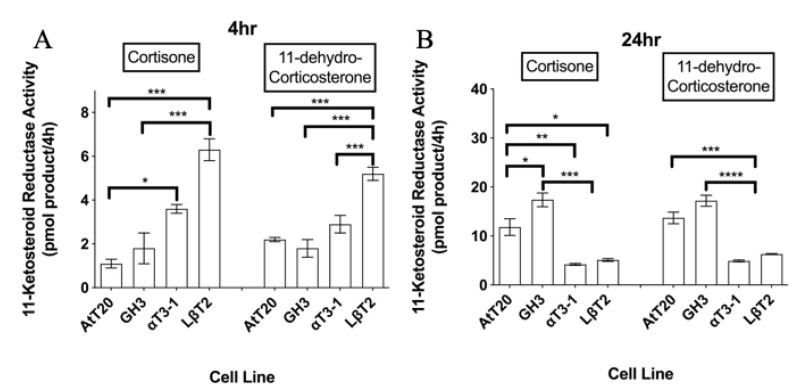
11-ketosteroid reductase activities in four pituitary cell lines measured over (**A**) 4 h or (**B**) 24 h using 100 nM [^3^H]-cortisone and 100 nM [^3^H]-11-dehydrocorticosterone as enzyme substrates. Data shown are mean ± SEM for three different cell passages pooled together (n = 3) (with triplicate measures in each passage). For a given time point and substrate, mean values with different superscripts differ significantly (*p* < 0.05) (one-way ANOVA with the Tukey–Kramer multiple comparison as the post hoc test).

**Figure 3 biology-08-00081-f003:**
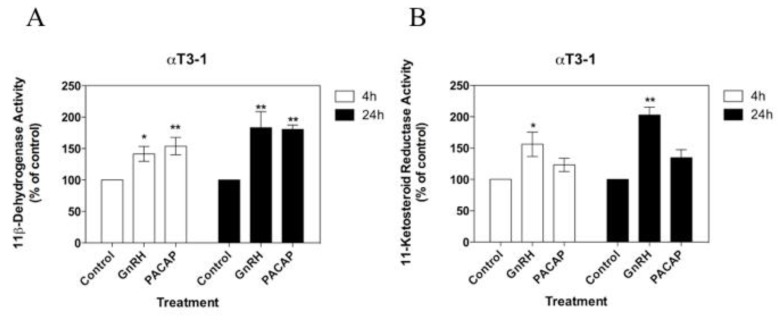
Effects of GnRH and PACAP on (**A**) 11β-dehydrogenase and (**B**) 11-ketosteroid reductase activities in αT3-1 cells over 4 h and 24 h using 100 nM [^3^H]-cortisol and 100 nM [^3^H]-cortisone as enzyme substrates. Values are mean ± SEM for three different cell passages (with triplicate measures in each passage). For each time point and enzyme activity, **p* < 0.05 and ** *p* < 0.01 versus corresponding control. Data shown are normalised as a percentage of the corresponding control enzyme activity in the absence of treatments (ranging from 3.1 to 4.1 pmol product/4 h for 11β-dehydrogenase activity, and from 3.0 to 4.6 pmol product/4 h for 11-ketosteroid reductase activity). Non-normalised data were analysed by one-way ANOVA with the Dunnet’s multiple comparison as the post hoc test).

**Figure 4 biology-08-00081-f004:**
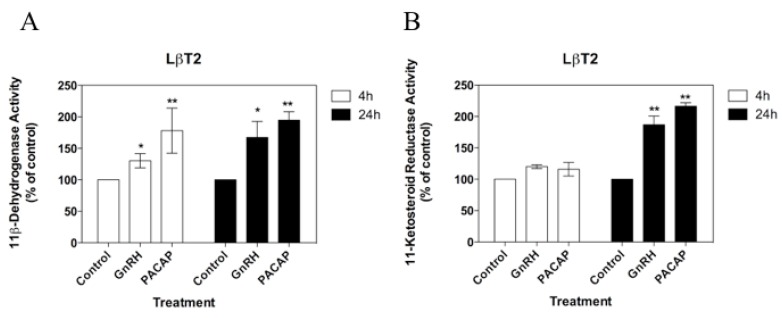
Effects of GnRH and PACAP on (**A**) 11β-dehydrogenase and (**B**) 11-ketosteroid reductase activities in LβT2 cells over 4 h and 24 h using 100 nM [^3^H]-cortisol and 100 nM [^3^H]-cortisone as enzyme substrates. Values are mean ± SEM for three different cell passages (with triplicate measures in each passage). For each time point and enzyme activity, * *p* < 0.05 and ** *p* < 0.01 versus corresponding control. Data shown are normalised as a percentage of the corresponding control enzyme activity in the absence of treatments (ranging from 2.4 to 4.5 pmol product/4 h for 11β-dehydrogenase activity, and from 5.4 to 7.2 pmol product/4 h for 11-ketosteroid reductase activity). Non-normalised data were analysed by one-way ANOVA with the Dunnet’s multiple comparison as the post hoc test).

**Table 1 biology-08-00081-t001:** Overview of the effects of GnRH and PACAP on the 11β-dehydrogenase (11βDH) and 11-ketosteroid reductase (11KSR) activities over 4 and 24 h in αT3-1 and LβT2 cells. (+ denotes significant stimulation of enzyme activity; ns denotes no significant effect on enzyme activity).

Endocrine Stimulus	Incubation of αT3-1 Cells for:				Incubation of LβT2 Cells for:			
	4 h		24 h		4 h		24 h	
	11βDH	11KSR	11βDH	11KSR	11βDH	11KSR	11βDH	11KSR
GnRH	+	+	+	+	ns	ns	+	+
PACAP	+	ns	+	ns	+	ns	+	+

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
