# Peer review of "Hormonal Regulation of Glucocorticoid Inactivation and Reactivation in αT3-1 and LβT2 Gonadotroph Cells"

_biology, 2019, doi:10.3390/biology8040081_

Round 1
Reviewer 1 Report
This paper addresses the regulation of bioavailable glucocorticoids/glucocorticoid metabolism in pituitary cell lines. There is documented evidence that elevated glucocorticoids induce infertility and the mechanism of how this might work at the level of the pituitary is an important question. The data show all pituitary cell lines can interconvert active and less active glucocorticoids, although the time course differed between cell types. Interestingly, GnRH and PACAP are able of enhancing the interconversion in gonadotrope cell lines. The data collected are sound and the paper is very well written. The following comments are offered:
Although it may not be feasible in this current study, it would be important to see if primary pituitary cells behaved similarly to the cell lines.
To support the conclusion that HSD11B1 is the main isoenzyme responsible for the effect seen, a knockdown of HSD11B2, with repeated treatments, would be helpful. However, since this point was only brought up in the discussion and doesn’t appear in other interpretation or the abstract, it would not be essential to add this and the “almost certainly” phrase appearing in the discussion (lines 209-210) could be toned down.
The authors mention that one difference between the gonadotroph, compared to the corticotroph and somatotroph, cell lines might be differential induction of enzyme expression (lines 212-223). Can the authors show this experimentally? It would add an important piece to the manuscript.
The author contributions statement seems to be the generic statement from the website. It should be filled in to include the specific author contributions to this paper.
Author Response
We are very grateful to the reviewer for their extremely positive comments regarding our manuscript and thank them for their suggestions. We respond to these comments below (in bold).
"Although it may not be feasible in this current study, it would be important to see if primary pituitary cells behaved similarly to the cell lines." We agree that it will be important to examine 11bHSD enzyme activity in non-immortalised pituitary cells and tissue. However, there are several benefits to the approach that we have taken, in using well-characterized, immortalised cell lines. αT3-1, LβT2, GH3 and AtT20 have been used for several decades as useful models of gonadotrope, somatolactotrope and corticotrope cells of the anterior pituitary. Our study enables us to examine 11bHSD activity in a single cell type; a heterogeneous population of primary pituitary cells does not enable this sort of investigation. Furthermore, enzymatic dispersion techniques required to make primary cultures are known to damage cell surface receptors, which would compromise our ability to use hypothalamic releasing factors such as GnRH and PACAP, to investigate the potential regulation of 11bHSD activity. Despite this, only in vivo approaches, such as the generation of pituitary-specific 11bHSD1/2 knock-out mice, will establish the contribution of the different pituitary cell types to 11bHSD activity in vivo.
"To support the conclusion that HSD11B1 is the main isoenzyme responsible for the effect seen, a knockdown of HSD11B2, with repeated treatments, would be helpful. However, since this point was only brought up in the discussion and doesn’t appear in other interpretation or the abstract, it would not be essential to add this and the “almost certainly” phrase appearing in the discussion (lines 209-210) could be toned down." Again, we agree with the reviewer and have toned down the language used to justify our interpretation of our data being indicative of 11bHSD1 activity. We have also added further justification to support our assertion that our data likely represent 11bHSD1 activity (lines 214-225).
"The authors mention that one difference between the gonadotroph, compared to the corticotroph and somatotroph, cell lines might be differential induction of enzyme expression (lines 212-223). Can the authors show this experimentally? It would add an important piece to the manuscript." There are multiple examples in the literature that describe a role for glucocorticoids in regulating 11bHSD expression, and we have cited these in the original manuscript (refs 32-44). To adequately investigate the effect of these ligands on 11bHSD expression in GH3 and AtT20 would require a significant experimental undertaking, and is beyond the scope of our cell biochemistry study. However, we very much agree with the reviewer that the potential mechanism by which glucocorticoids may upregulate 11bHSD expression in these cells should be followed up in due course.
"The author contributions statement seems to be the generic statement from the website. It should be filled in to include the specific author contributions to this paper." Thank you for pointing this out, and many apologies for this typographical error - we have corrected this in the revised manuscript.
Reviewer 2 Report
In this manuscript, Michael et al. describe glucocorticoid effects on gonadotroph cells. Overall, the manuscirpt is easy to digest. It is brief, but on-point, and appears well written. At face-value, references, results, and conclusions seem appropriate.
I have only minor comments, which I think if addressed will improve the manuscript.
First, the statistical denotation of effects are difficult to interpret. The various superscripts are not intuitive. I implore the authors to consider a standard * approach, with # for trends.
Second, more detail in figure legends will help naive readers consider the contents of experiments and their significance.
Third, I would like more clarification regarding numbers. The authors do report these in word-format. I would like to see these more transparently spelled out in figure legends (e.g., "x observations from x total replicates derived from x lines). The authors can develop their own format, but it is important that n's are easily found and digestible.
Fourth, two correction methods are listed. It would be to the authors strength to provide a statistical rationale for their use and adaptation. For example why were two used? Is this a design matter? Is this a problem with variability or observed power? This may seem a daunting challenge to resolve for the statistically naive, but it is not, and it is important that authors describe their design and analysis parameters to ensure they are justified. This will help readers devolve the robustness of effects, and subsequent replication parameters.
Fifth, should the authors consider alternative assessments to confirm critical data (even if at one time-point as proof of concept, or one specific target if radiometric conversion was shown to be specific to a target) such as enzyme zymography for activity differences or differences in detection methods for hormone products? This should perhaps be done anyway for robustness, however if there is a compelling reason not to do this then the authors may make their point.
Sixth, in Figure 4 the ctrl group has no SEMs. This is odd. Was only one control used? There should be a SEM for this group if multiple ctrls were used, as they would be averaged, then each value normalized back to the group average, to yield SEMs. Thus, please consider reporting how data are normalized and analyzed for transparency.
Seventh, can nM concentrations be substantiated for their selection?
Overall, these are honestly very minor points that can be easily addressed by the authors, resulting in a more transparent and robust manuscript. I therefore only request minor revision, and trust that the reviewers will resolve these questions/matters.
Author Response
Again, we a re most appreciative of the efforts that the reviewer has gone to in assessing our manuscript, and have responded to their queries and suggestions below (in bold).
"First, the statistical denotation of effects are difficult to interpret. The various superscripts are not intuitive. I implore the authors to consider a standard * approach, with # for trends." Thank you for this suggestion - we have now adjusted the graphs in Figs 1A, 1B, 2A and 2B accordingly, and hope that this improves clarity.
"Second, more detail in figure legends will help naive readers consider the contents of experiments and their significance." We thank the reviewer for this suggestion, and agree that our initial figure legends were a little too brief. Consequently, we have added more detail to each of the figure legends to better describe how data have been presented.
"Third, I would like more clarification regarding numbers. The authors do report these in word-format. I would like to see these more transparently spelled out in figure legends (e.g., "x observations from x total replicates derived from x lines). The authors can develop their own format, but it is important that n's are easily found and digestible." We apologise to the reviewer for not making this important information easier to access in our original submission, and have now inserted additional descriptions in the Materials and Methods (line 130-131) as well as in each figure legend.
"Fourth, two correction methods are listed. It would be to the authors strength to provide a statistical rationale for their use and adaptation. For example why were two used? Is this a design matter? Is this a problem with variability or observed power? This may seem a daunting challenge to resolve for the statistically naive, but it is not, and it is important that authors describe their design and analysis parameters to ensure they are justified. This will help readers devolve the robustness of effects, and subsequent replication parameters." Thank you for raising this; however, we have only employed one correction (or 'normalisation') approach in our current study, and have now expanded the description, when we show '% of control' data. Furthermore, we now report the range of non-normalised data values in each of these experiments (within the figure legends), to give a reflection of the variability of enzyme activity in each of the three individual experiments.
"Fifth, should the authors consider alternative assessments to confirm critical data (even if at one time-point as proof of concept, or one specific target if radiometric conversion was shown to be specific to a target) such as enzyme zymography for activity differences or differences in detection methods for hormone products? This should perhaps be done anyway for robustness, however if there is a compelling reason not to do this then the authors may make their point." The use of radiolabelled steroids represents the gold standard approach for the assessment of steroid interconversion; utilising HPLC, instead of TLC, would only provide an incremental increase in resolution, but would essentially be the same experimental approach. Alternative methodologies might be useful to compare with our radiometric conversion assays, but suffer from the confounding issue of how to discriminate between endogenously produced/metabolised steroids and those provided exogenously. The approach we have taken allows us to directly determine the enzyme activity as % conversion.
"Sixth, in Figure 4 the ctrl group has no SEMs. This is odd. Was only one control used? There should be a SEM for this group if multiple ctrls were used, as they would be averaged, then each value normalized back to the group average, to yield SEMs. Thus, please consider reporting how data are normalized and analyzed for transparency." As mentioned above, we have provided further information as to how we have normalised data. There are three values for every experimental group - the reason for the lack of SEM on the control group is that the data are normalised as a function of that value within in each experiment - so, by definition, the control group is '100%' in each experiment.
"Seventh, can nM concentrations be substantiated for their selection?" nM concentrations are certainly physiological relevant (for example, in humans a range of 150-600nM would be expected); furthermore, the concentrations we have used fir the Km's for the enzymes.
Round 2
Reviewer 2 Report
The reviewer is satisfied that the reviewers made reasonable attempt to improve the manuscript.